# Current Advances of Nanomaterial-Based Oral Drug Delivery for Colorectal Cancer Treatment

**DOI:** 10.3390/nano14070557

**Published:** 2024-03-22

**Authors:** Nuoya Wang, Liqing Chen, Wei Huang, Zhonggao Gao, Mingji Jin

**Affiliations:** 1State Key Laboratory of Bioactive Substance and Function of Natural Medicines, Institute of Materia Medica, Chinese Academy of Medical Sciences and Peking Union Medical College, Beijing 100050, China; wangnuoya@zgyxkxyywyjs.wecom.work (N.W.); chenliqing@imm.ac.cn (L.C.); huangwei@imm.ac.cn (W.H.); 2Beijing Key Laboratory of Drug Delivery Technology and Novel Formulations, Institute of Materia Medica, Chinese Academy of Medical Sciences and Peking Union Medical College, Beijing 100050, China; 3Department of Pharmacy, Yanbian University, Yanji 133000, China

**Keywords:** oral, nano-drug delivery system, oral colon-targeted drug delivery system (OCDDS), colorectal cancer

## Abstract

Colorectal cancer (CRC) is a common malignant tumor, and traditional treatments include surgical resection and radiotherapy. However, local recurrence, distal metastasis, and intestinal obstruction are significant problems. Oral nano-formulation is a promising treatment strategy for CRC. This study introduces physiological and environmental factors, the main challenges of CRC treatment, and the need for a novel oral colon-targeted drug delivery system (OCDDS). This study reviews the research progress of controlled-release, responsive, magnetic, targeted, and other oral nano-formulations in the direction of CRC treatment, in addition to the advantages of oral colon-targeted nano-formulations and concerns about the oral delivery of related therapeutic agents to inspire related research.

## 1. Introduction

Colorectal cancer (CRC) is a common malignant tumor of the digestive system, with at least 1–2 million cases reported annually. It is the third most common cancer, with the fourth highest death rate after breast, lung, and liver cancers worldwide [1]. CRC is a malignant tumor of the glandular and epithelial cells of the colon or rectum that originates from polyps in the epithelial lining of the colon and rectum, invades the muscles and nearby lymph nodes, progresses to CRC, and spreads to the liver, lungs, and other organs [2]. Traditional treatments for CRC include surgical resection and radiotherapy. However, problems, such as local recurrence, distal metastasis, and intestinal obstruction complications, persist.

With the development of nanomedicine, drug delivery systems (DDS) have become widely used in treating CRC, showing great potential application value and development prospects. Nano-DDS can improve therapeutic drug retention, accumulation, penetration, and targeted cell uptake at the tumor site, achieve controlled release in the tumor extracellular matrix or tumor cells, and improve the efficiency of regulating anti-tumor immune response and the therapeutic effect. Oral administration is a safe and convenient method of drug delivery. Oral administration of nanoscale drugs can increase drug bioavailability, improve drug stability, and achieve targeted drug delivery.

The oral colon-targeted drug delivery system (OCDDS) is a drug delivery strategy designed to deliver therapeutic agents to the site of colonic disease to increase the efficacy of drug therapy, improve bioavailability, reduce systemic side effects, and facilitate the treatment of CRC and inflammatory bowel disease (IBD). Despite the advantages of OCDDS for treating CRC, it still faces many obstacles due to the complex physiological barriers in the human body. The intestinal mucus layer of the human body rapidly traps and eliminates external particles via adhesion and spatial barriers, preventing pathogens and various toxins from invading mucus epithelial cells while restricting the effective diffusion and absorption of drugs, which is the biggest bottleneck for DDS for CRC. Oral nano-delivery systems modified with different materials can allow the drug to penetrate the intestinal mucus barrier easily while protecting it from the acidic environment in the gastrointestinal tract (GIT) or the immune system [3,4]. Xu et al. [5] encapsulated insulin in poly(lactide-co-glycoside) (PLGA) nanoparticles with a folic acid-chitosan modification to overcome the negatively charged PLGA nanoparticles to penetrate the mucus barrier and improve insulin bioavailability. Farida E. E et al. [6] developed a PLGA-loaded nano-curcumin to explore the drug’s effect on certain inflammatory markers in colon cancer in mice. Nano-curcumin prepared with PLGA was shown to modulate the levels of TNF-α, IL6, and vascular endothelial growth factor, and exhibited better in vivo long-circulation effects and tissue permeability in colon histology studies. This simple modification of the drug could inform future treatments for colon inflammation and cancer. Shadab et al. [7] synthesized pH-responsive nanohydrogels with a protein–polysaccharide structure. The oral dual-layer system decreases the gastric residency time and increase the intestinal-colonic residency with NAR enhanced dissolution profile using the soy protein complexation. It effectively avoids recognition and phagocytosis by the immune system, sustained release, and pH selectivity, showing better cytotoxicity in CRC.

## 2. Physiological Environmental Factors and Major Challenges for CRC Therapy

The efficacy of oral colon-targeted systems is contingent upon overcoming physiological impediments within the GIT. These obstacles encompass variations in pH conditions, transit time, fluid volume, enzymatic content, microbiota diversity, and mucus layer (Figure 1). Alimentary content and inter-/intra-individual variability constitute additional variables impacting the optimal drug delivery to the colon. These factors should also be considered when formulating colon-targeted systems [8]. This section discusses some of these GIT barriers mentioned above.

### 2.1. Intestinal pH

The GIT is typically segmented into three primary regions: stomach, small intestine (duodenum, jejunum, and ileum), and large intestine (colon, cecum, and rectum). An initial challenge in oral drug administration pertains to the diverse pH profiles across the GIT. The stomach harbors an acidic environment with a pH ranging from 1 to 3, transitioning to pH 6 in the duodenum, escalating to pH 7.4 in the terminal ileum, and ultimately settling between pH 6 and 8 in the colon [9]. Consequently, designing delivery systems targeting the colon could exploit these pH disparities within the GIT as a strategy for precise drug delivery. However, GIT pH levels vary considerably among individuals due to factors like dietary intake, microbial metabolism, and the active or remissive state of diseases. This variability can impact the efficacy of pH-sensitive systems, thereby affecting drug release at the intended site.

### 2.2. Transit Time

Similarly, transit time is another factor affecting oral delivery and drug bioavailability in the colon. The normal transit time in the small intestine is approximately 4 h, with an inter-individual variation of 2–6 h; however, the colon varies significantly, ranging from 6 to 70 h. It is important to mention that factors, such as type of diet, fasted/fed state, mobility, stress, and colon diseases, influence the GIT transit time. For instance, patients with active Ulcerative Colitis (UC) have colonic transit time faster than healthy individuals. Diarrheal events, common in patients with IBD and CRC, accelerate the peristaltic movement of the GIT and reduce transit time. Additionally, the size of the dosage forms affects the transit time, with larger particles having shorter transit times than smaller particles [10]. The reduction in GIT transit time influences the retention/clearance of DDS administered orally, which can reduce their therapeutic effect at the action site [11]. Accordingly, the GIT transit time becomes a crucial factor to be considered when developing time-dependent, as well as micro- and nanoparticulate systems.

### 2.3. Colonic Fluid

The average colonic fluid volume was 13 mL, ranging from 1 to 44 mL in the fasted state [12]. The presence of food alters the colonic fluid volume and digestive enzyme activity, influencing nutrient absorption and carbohydrate/polysaccharide breakdown. Conversely, patients with CRC experience heightened fluid secretion and reduced water reabsorption, diluting colonic volume. Body fluid volume influences the disintegration and dissolution rate of orally administered drugs. Therefore, a decreased colonic fluid volume affects the dissolution and absorption of a drug, thereby affecting its bioavailability in the colon.

### 2.4. Enzymatic Content

Various enzymes, such as lysozymes, glucuronidases, glycosidases, aldehyde dehydrogenase (ALDHs), and specific proteases, are present in the colon. ALDH function in cancer stem cells is not limited to drug detoxification, as the genetic suppression of individual isoforms reduces tumorigenicity and metastasis. Feng et al. [13] pioneered lead ALDHs isoform-selective guanidino antagonists (IGUANAs) with proteome-wide targeting specificity to selectively block the growth of colon cancer spheroids and organ tissues. It acted as a better sensitizer in colon cancer cell lines that had developed resistance to the action of 5-FU, effectively inhibiting globular growth. They examined its effects on primary human organoid cultures representing normal colon, colon adenocarcinoma, and rectal adenocarcinoma, and found that it could inhibit the growth of each line. Lysyl oxidase (Lox) is a copper-dependent enzyme encoded by Lox gene [14,15]. Clinical studies have investigated the association between Lox expression and clinicopathological features, progression, and prognosis of GI cancers, including colorectal, esophageal, Hepatocellular Carcinoma (HCC), gastric, and pancreatic carcinoma. Lox expression is significantly higher in colorectal tumors than in other tissues, and patients with high Lox expression have poorer prognosis [16]. Consequently, Lox inhibitors may have a strong potential to reduce cancer progression alone or in combination with chemotherapy drugs against GI malignancies.

### 2.5. Colonic Microbiota

In 1995, a study reported 15 bacterial species associated with a higher risk of developing CRC. These included two Bacteroides species (*B. vulgatus* and *B. stercoris*), two Bifidobacterium species (*B. longum* and *B. angulatum*), five Eubacterium species (*E. rectale 1* and *2*, *E. eligens 1* and *2*, *E. cylindroides*), three Ruminococcus species (*R. torques*, *R. albus* and *R. gnavus*), *Streptococcus hansenii*, *Fusobacterium prausnitzii*, and *Peptostreptococcus productus* [17]. *Fusobacterium*, *Porphyromonas*, *Peptostreptococcus*, *Gemella*, *Mogibacterium*, and *Klebsiella* were enriched in patients with CRC, whereas *Feacalibacterium*, *Blautia*, *Lachnospira*, *Bifidobacterium*, and *Anaerostipes* were reduced [18]. Moreover, the microbiota of cancerous tissues exhibits a lower diversity than that of non-cancerous normal tissues [18].

This high enzymatic and microbiota activity in the colon is a key strategy for designing colon-targeted DDS. Systems based on natural polysaccharides (guar gum, starch, and pectin) have been widely exploited, with several examples of drug delivery to the colon due to their specific degradability by colonic enzymes from anaerobic bacteria [19]. Additionally, some polysaccharides can act as prebiotics (non-digestible food components fermented by colonic microbiota) to colonic bacteria. Accordingly, they can stimulate the proliferation of bacteria and modulate the colonic microbiota. Prodrug systems also exploit colonic enzymes to convert inactive molecules into pharmacologically active molecules for colon-targeted delivery [20]. However, some conditions, such as diet type, drug therapy (antibiotics and laxatives), and disease state, can cause changes in enzyme secretion, resulting in dysbiosis (change in microbial composition) and decreased intestinal repair [21]. These colonic microbiota alterations can affect systems based on polysaccharides, prodrugs, and the drug release mechanism.

### 2.6. Mucus

The colonic mucus (CM) layer consists of glycosylated mucin formed from substances secreted by the cup cells of the colonic epithelium and is approximately 110~160 μm thick [22]. The inner layer is tight and cannot be penetrated by microorganisms. The outer layer is loose and prevents the luminal epithelium from being adsorbed by foreign bodies swallowed from the mouth as they pass through the intestines and attack the inner layers of the intestines. The mucus barrier integrity is important to maintain daily intestinal homeostasis and prevent bacteria from damaging the gut [23]. However, it can be affected by several factors, including a lack of certain essential substances in the gut. The gut symbiotic microorganisms will “munch” the mucus in the gut. Damage to CM can deteriorate the colonic barrier, allowing bacteria to enter into epithelial cells and causing inflammatory reactions and cancer. Since mucus is continuously secreted, it reduces the retention time of substances in the CM, thereby removing harmful substances and therapeutically useful drugs. Therefore, mucus hinders the absorption/penetration of orally administered drugs via the intestinal epithelium, limiting their therapeutic efficacy in the colon. The substances commonly used for CM penetration include hyaluronic acid (HA) [24,25], alginate [26], and other polysaccharides.

### 2.7. The Gut-Associated Lymphoid Tissue (GALT)

The gut-associated lymphoid tissue (GALT) is one of the most widespread immune organs, which accounts for 70% of the body’s lymphatic system [27]. Drugs taken orally are absorbed and transported through the GALT, which is located in the submucosal layer of the intestinal tract and contains a large number of isolated lymphoid follicles (ILF) and immune cells capable of recognizing and ingesting orally ingested antigens and drugs [28]. When oral drugs enter the intestine, GALT absorbs them and transports them to the systemic circulatory system for therapeutic effects. M cells in GALT appear to be the primary route of entry for several human pathogens such as *Salmonella*, *Yersinia*, *Mycobacteria*, and *prions*, and oral formulations targeting these pathogens would be more effective than parenteral formulations. The diagnosis of Crohn’s disease (CD) and early CRC can be confirmed by looking at inflammatory lesions in the GALT. For example, increased numbers of lymphocytes have been reported in the ILF of patients with CD, suggesting that the ILF are a site of adaptive immune activation in patients with CD [29]. Therefore, GALT could be a target site for the treatment of CRC, and it is also important to consider how orally administered drugs escape T cells and M cells in the immune zone.

## 3. Need and Design Strategies for Novel OCDDS

CRC results from transforming normal colonic epithelial cells into adenomatous polyps, and most chemotherapy methods are currently used. Intravenous injections can cause trauma and toxic side effects in normal tissues. If colon cancer treatment drugs are prepared into oral-targeted formulations and delivered specifically to the colon, it can help reduce systemic toxicity and improve drug bioavailability. OCDDS has recently attracted more attention due to its targeted delivery of drugs to target tissues, low toxicity, and high patient compliance. However, it is difficult to selectively increase the drug concentration at the lesion site and obtain the expected effect due to the poor absorption of oral chemotherapy drugs, instability in the GIT, and easy degradation and inactivation during internal circulation. The construction of diversified nano-delivery systems can improve the stability and targeting of drugs via functional materials and improve the anti-tumor efficacy of oral chemotherapy drugs.

### Systemic Issues with DDS

In addition to overcoming the physiological environment of the GIT, novel OCDDS must focus on improving drug efficacy, targeting, and survival. Researchers are developing innovative approaches, such as nanotechnology-based drug carrier systems, to enhance the physicochemical and pharmacodynamic properties of chemotherapeutic drugs or the targeted delivery of drugs to various types of tumor tissues to optimize therapeutic efficacy while reducing off-target effects. Nanoparticles are promising therapeutic modalities for CRC and other cancers because their properties are beneficial for addressing the problem of difficult systemic drug delivery. The European Commission proposes that the only determining factor in deciding whether a material is also a nanomaterial is the distribution of particle sizes contained in the material, and that a material is a nanomaterial if more than 50 per cent of the particles are between 1 nm and 100 nm in size, i.e., if it is above the minimum threshold. The drug loading capacity can be increased by core encapsulation and surface adsorption of nanocarriers. Intelligently designed nanoparticles allow for particle size control, which improves nanoparticle uniformity while avoiding the challenges of surface charge aggregation due to irregular shapes. Due to their ultra-small size, dispersion and super encapsulation, they are able to prolong in vivo circulation time, thereby reducing clearance from the renal system [21]. Additionally, the physical confinement of diagnostic reagents and anticancer drugs improves their pharmacokinetic properties while reducing the number of administrations, due to the nanoparticles allowing for continuous or controlled release. This controlled release circumvents multi-drug resistance as cells actively take up significant amounts of the drug intracellularly. Nanoparticles promote anticancer mobility in somatic circulation, increase cellular uptake, and provide benefits superior to bulk equivalents. Another advantage of using nanoparticles is that they can tailor their surfaces using targeted ligands (peptides or small-molecule ligands) to increase their selectivity for desired cells/tissues while decreasing the availability of chemotherapeutic drugs in normal tissues. Amphiphilic lipid-based nanoparticles, such as micelles, have hydrophilic surfaces and hydrophobic cores that can increase the hydrophobic drug solubility. In addition, the use of fluorescent dyes as an alternative strategy to synthesize hydrophilic nanomaterials for the delivery of colorectal cancer treatment drugs helps to visualize the delivery of drugs to colorectal tumor sites during imaging, so as to treat colorectal cancer diseases more effectively. Quantum dots, nano-shells, carbon nanoparticles, nanobodies, and paramagnetic nanoparticles have structural properties that can be used for imaging and diagnostics [30].

This review categorizes oral nanomaterials based on their mechanisms of action and examines the research progress of controlled release, microenvironment-responsive, magnetic, targeted, and adsorption-based oral nanomaterials in treating CRC to inspire related research.

## 4. Controlled-Release Oral Nano-Formulations

Controlled-release oral nano-formulations use nanotechnology to encapsulate drugs in nanoscale carriers to control the release rate and targeted drug delivery in the intestinal tract, thereby enhancing drug efficacy. It enhances drug stability, bioavailability, sustained-release time, targeting, stability, solubility, and patient compliance.

### 4.1. Multilayered Types of Controlled Release Nano-Formulations

The liposomes of multilayered formulations are mainly composed of phospholipids. Phospholipids can self-assemble into bilayers or multilayered vesicles in an aqueous environment, with a thickness of about 4–5 nm [31]. In this process, water-soluble drugs interact with the center of the phospholipid bilayer to dissolve them in the aqueous phase. Lipid-soluble drug molecules interact with the hydrophobic tails of liposomes to disperse drugs inside the phospholipid bilayer. The surface of liposomes is modified with natural or synthetic multimers to form a hydrophilic protective layer that enhances their stability and circulation time. This “stealth” effect is produced by inducing protein binding and regulation as well as reducing degradation using metabolic enzymes. This is due to the fact that certain polymers block the entry or recognition of blood proteins, protein hydrolases, and antibodies during blood circulation [32]. Such an assembly structure will enable it to have a higher drug loading capacity when encapsulating the drug, and a certain thickness of the shell will enable it to pass through the gastrointestinal tract without leaking the drug and gradually release the drug when it reaches the target site in the intestine, so as to achieve the effect of controlled release. Song et al. [33] constructed cyclic arginylglycylaspartic acid (cRGD) peptide-modified multilayered liposomal DDS for targeted oral apatinib administration. The comparison between intravenous and gavage administration revealed that the latter had stronger fluorescent signals in vivo and greater accumulation at the tumor site in the mouse model. The ability of polyethylene glycol (PEG)-modified liposomes to show a longer in vivo circulation time is due to enhanced vascular permeability at the tumor site (EPR effect), thus enhancing drug accumulation in solid tumors. Yu et al. [34] constructed a polyethylene glycolyzed apatinib liposome using MPEG-PCL(Lipo-Apa). This multilayered liposome is an effective DDS, and its amphiphilic nature increases its permeability and retention time in tumor tissues and delays the release of highly concentrated drugs in tumor tissues to achieve a locally controlled release effect. In a subcutaneous xenograft and peritoneal metastasis model of CRC, the combination therapy of gavage Lipo-Apa showed significant anti-tumor activity, a slow and controlled release effect, and a reduction in tumor angiogenesis.

### 4.2. Porous Types of Nanomaterials

Mesoporous silica nanoparticles (MSNs) are porous three-dimensional carbon-based nanomaterials with pore sizes ranging from 2–50 nm, which have good biocompatibility, high specific surface area, and controllable mesoporous structure [35,36], making them the most promising biomolecule carriers for nanotechnology and drug-targeted delivery [37]. MSNs exhibit low toxicity and easy transmembrane transport, are easy to swallow, and have promising applications in tumor therapy [38,39]. Wang J et al. [40] designed a colon-targeted nanodrug delivery platform for oral administration using small molecule peptides (M27-39) and folic acid (FA) modified MSNs. M27-39@FA-MCNs showed smooth changes in the capillary condensation stage of the N_2_ sorption isotherms, without hysteresis loops, and showed remarkably fewer mesoporous structural characteristics than blank FA-MCNs. These features make it possible to prevent nanoparticle rupture and drug leakage while administering the drug orally and by programmatically releasing the drug through the mesoporous tissues after reaching the target site, yielding superior sustained and controlled release performance. Bioactive calcium phosphate ceramics (CaPs) are a recently discovered novel inorganic material with a porous structure that can be modified with active substances to be used as drug carriers to control the release of active substances. Dagmara Słota et al. [41] modified CaPs with more clindamycin loading by physisorption. Drug adsorption on ceramic powders is mainly accomplished by forming bonds between Ca^2+^ ions on the ceramic surface and oxygen atoms in the drug molecule. This is because the larger pore surface area leads to more efficient migration of the drug into the interior of the ceramic, resulting in later and longer drug release. This study shows that the crystallinity of the tested nano-powder also affects the loading capacity of the active substance as well as the amount of drug released. This is a great inspiration for orally administered pharmaceuticals, where sturdy bone-like structures might be more acid-resistant, favoring the passage of drugs through the acidic environment of the gastrointestinal tract to reach the intestines for treatment.

## 5. Microenvironment-Responsive Oral Nanomaterials

Microenvironment-responsive oral nano-formulations are usually composed of nanocarriers, such as nanoparticles, nano-capsules, or nanogels, that respond to specific physiological environments or internal and external stimuli (pH, solubility, temperature, and light) to achieve controlled drug release.

### 5.1. pH-Responsive Oral Nano-Formulations

The pH is a commonly used delivery response element for the precise localization of organs (vagina and GIT) or organelles (lysosomes and Golgi apparatus). It has also been used to release drug fractions under specific pathological conditions, such as cancer, inflammation, or ischemia, which are associated with significant pH changes [42]. Colon-targeted polymers should tolerate gastric acid and the pH of the proximal small intestine and dissolve in the pH of the terminal ileum and colon. Accordingly, a pH-dependent DDS with a dissolution threshold of 6.0–7.0 is expected to delay drug dissolution and prevent unwanted drug release in the GIT before reaching the colon [43]. This allowed for the slow-release properties of the drug formulations. FA-conjugated nanoparticles (FNPs) developed by Abbasi [44] were delivered to the colon via pH-sensitive hydrogels synthesized using free-radical polymerization to provide sustained drug release upon reaching the colonic site at pH 7.4, which is a major challenge for most hydrophobic drugs that cannot be delivered orally to the colon. Tian et al. [45] developed chitosan quaternary ammonium-modified pH-responsive butadienolide nanocrystals that protect butadienolide terpene lactones from damage in the acidic environment of the stomach, enhance drug enrichment in the colon to achieve synergistic release, significantly increase oral bioavailability, and exert therapeutic effects in treating CRC.

Although the pH response is widely used in smart DDS, there are limitations to using pH as a trigger in the tumor microenvironment. Amino acid abnormalities can be detected in serum, plasma, and tumor tissues [46], and acidic tissue fluids are usually far from the blood stream, thus resulting in inaccurate delivery of nanoparticles [47]. Additionally, pH changes between healthy and tumor tissues do not differ significantly [48,49]. Therefore, they should be combined with different response elements (temperature or enzymatic reactions) to ensure precise release at the target site. Song et al. [50] developed pH-conjugated enzyme-responsive dual-responsive nanoparticles (Gd-MHAPNPs) to achieve precise targeted drug delivery (Figure 2).

### 5.2. Enzyme-Responsive Oral Nano-Formulations

Enzymes secreted by the colonic flora act as specific stimuli to trigger colonic drug release and have been used to aid drug delivery to cancer cells [51]. Among the various enzymes, proteases are particularly relevant for developing novel DDS, as they are commonly overexpressed in diseases, such as cancer and inflammation. Trypsin [51] is a key digestive protease that plays a key role in regulating pancreatic exocytosis, affecting the release of many other digestive enzymes [52]. Matrix metalloproteinases (MMPs) are zinc-dependent endopeptidases known for their involvement in cancer prognosis [53] and have been widely explored for drug delivery and imaging [54]. According to a clinical statistical trial in a Chinese population, MMP variants are potential genetic markers for CRC risk [55]. Rashidzadeh et al. [56] prepared enzyme-responsive hydrogels containing methotrexate (MTX) and chloroquine (CQ) in polymethylmethacrylic acid (PMAA), catalyzing the drug release in the presence of proteolytic enzymes. Oral administration of this drug was biocompatible and effective in inhibiting the growth and proliferation of CRC cells. Li et al. [57] developed an effective enzyme-triggered controlled release system (Cur-CD-CANPs) using a curcumin–cyclodextrin (CD–Cur) inclusion complex as the core and low molecular weight chitosan and unsaturated alginate resulting nanoparticles (CANPs) as the shell. It exhibited pH-sensitive and α-amylase-responsive release characteristics. Cur was rapidly released from this system with α-amylase, reaching 90% within 4 h. This drastic release in the presence of α-amylase was due to β-CD enzymatic degradation, chain scission in CD, and drug release. These results indicate that the formed NPs have pH sensitivity and enzyme-responsive release characteristics.

### 5.3. ROS-Responsive Oral Nano-Formulations

Reactive oxygen species (ROS) are primarily produced in the intestine by mucosal resident and immune cells [58]. Phagocytosis by leukocytes and neutrophils releases large amounts of ROS under inflammatory conditions. The colon has high expression of reduced nicotinamide adenine dinucleotide phosphate oxidase 1 (NOX1) [59], arresting the cellular differentiation cycle by triggering endoplasmic reticulum (ER) stress, thereby implicating tumorigenesis and progression [60]. Studies have revealed that mildly to moderately elevated ROS concentrations can promote cancer progression, whereas excess ROS mediates apoptosis and modulates the tumor microenvironment. Chen et al. [61] developed chitosan nanoparticles (CS NPs) encapsulating both photothermal (IR780) and photodynamic (5-aminolevulinic acid (5-ALA)) reagents for photothermally enhanced photodynamic therapy by non-invasive oral administration. Photodynamic generation of ROS improves the tumor site microenvironment, enhances the tumor-killing effect of the drug, and significantly enhances ROS-responsive release at the tumor site.

### 5.4. Photo-Responsive Oral Nano-Formulations

Light-responsive DDS employing photosensitized carriers displayed on/off drug-release mechanisms upon irradiation stimulation. Various wavelengths of light (ultraviolet (UV), near-infrared (NIR), and visible) have been reported and discussed for use in photo-responsive DDS. Visible and UV light are unsuitable for in vivo therapy due to their low penetration, while NIR is an ideal source of light for monitoring drug release due to its safety and better tissue penetration [62]. However, since its radiation intensity decreases very rapidly in deeper areas, its use in treatment protocols for any deeper tissues should be considered in detail. Various drug release mechanisms have been reported using this system, including the photothermal effect based on light conversion into heat by a photo-thermite that breaks down the nano-capsules to release the drug. Chen et al. [63] reported the development of nanostructured lipid carriers (NLCs) suitable for non-invasive oral delivery of a near-infrared photosensitizer dye IR780. Local laser irradiation activated the anti-tumor activity of IR780 while simultaneously generating a large amount of heat. Enhanced IR780 accumulation in the subcutaneous tumors of mice significantly affected the tumor growth rate, and this phenomenon was only related to photo-responsiveness. They created a safe and non-invasive method for orally administrating IR780@NLCs, enabling systemic tumor delivery of photosensitive dyes for photo-responsive therapy.

## 6. Magnetic Oral Nano-Formulations

Magnetic oral nanoparticles are oral drug formulations that utilize magnetic nanoparticles (MNPs) as carriers for drug localization, controlled release, and targeted delivery to the GI tract. MNPs are usually made of ferrite, magnetic materials, or iron oxide and are surface-modified with targeting ligands for specific targeted delivery (Figure 3). Commonly used MNPs (Figure 4) display various unique magnetic properties, which, if properly designed, can lead to specific advantages such as enhanced biocoupling sites and plasma half-life [64]. Nano-formulations based on MNPs have many advantages. 1. Targeting: MNPs can achieve targeted delivery of drugs via surface-modified targeting ligands, reducing their impact on non-targeted tissues and improving drug efficacy. 2. Controlled release: MNPs can realize controlled release of drugs via specific magnetic field stimulation, allowing the drugs to be released only when needed, improving drug stability and bioavailability. 3. Localization and navigation: MNPs can be guided by an external magnetic field to achieve drug localization in the digestive tract, allowing the drug to act directly on the target tissue or cells, thereby improving local drug efficacy. 4. Increased drug bioavailability: MNPs can improve the stability and controlled release of drugs, increase drug bioavailability, reduce drug metabolism and excretion, and improve drug efficacy. 5. Enhanced intestinal absorption: Magnetic nanoparticles can promote drug absorption in the intestinal tract and increase drug bioavailability, thus reducing drug dosage and side effects.

### 6.1. Magnetically Targeted Oral Nano-Formulations

An external magnetic field guides the localization of MNPs to the tumor site, increasing drug accumulation and action in the tumor tissue and improving therapeutic efficacy. Therefore, magnetic nano-preparations may be potential candidates for combined anti-tumor therapy in colon cancer. It is important to note that the strength of MNP magnetization strongly decreases with its size, which is a key factor to consider when applying this treatment method. DDS can precisely target the cancerous site by loading smaller sized Fe_3_O_4_ or modified iron oxide nanoparticles under the action of an external magnetic field at the colon cancer site. Based on these properties, a team prepared dimercaptosuccinic acid (DMSA)-coated Zn^2+^-doped magnetite nanoparticles, DMSA-Zn_0.4_Fe_2.6_O_4_, which maintained their original structure after passing through the gastrointestinal wall, entered the circulatory system via the small intestinal barrier, and reached the spleen and other body parts. It is well-accepted that magnetite nanoparticles can be transported to the spleen via intravenous injection. However, whether oral nanoparticles can be transported from the GIT to the spleen remains unclear, although they overcame this by coating it with DMSA. This preparation method has several advantages. First, it caused no significant damage to the mouse kidneys and spleens. Second, they have good blood compatibility. Third, it is magnetically targeted to the small intestinal site and penetrates the intestinal mucosa to target specific organs. Fourth, it retains its initial crystalline structure, a distinctive advantage for biomedical applications. Lu et al. [66] used superparamagnetic iron oxide nanoparticles (SPIONs) loaded with triglycerides of March laurel (LNP), prepared as lipid dextran microgels. This system uses alternating magnetic fields and dextran for layered dual targeting, resulting in colon retention and increased drug uptake by cancer cells. Oral delivery reduces gastrointestinal adhesion and prevents cellular transport of the drug via proton-coupled transporter proteins in the small intestine during oral delivery to the colon. The released drug recognizes and internalizes colon cancer cells upon transit to the colonic site, significantly inhibiting tumor growth in mice with in situ colon cancer and suppressing metastatic peritoneal carcinomas.

### 6.2. Magneto-Thermal Responsive Oral Nano-Formulations

The use of MNPs in response to an external alternating magnetic field produces a magneto-thermal effect, converting magnetic energy into thermal energy and localizing warming to kill cancer cells. Shen et al. [67] developed an NP-based chemo/magnetothermal combination therapy system for oral delivery and local treatment of colon cancer—DFSLNs. The present study demonstrated the effective accumulation and cellular uptake of doxorubicin and a superparamagnetic iron oxide nanoparticle-loaded solid lipid nanoparticle (SLN) delivery system for chemo/magnetothermal combination therapy in tumors by hierarchical targeting of FA and dextran-coated SLN surfaces in a sequential layer-by-layer manner. The most notable role of SPION—local combination treatment of DOX chemotherapy and hyperthermia therapy upon the activation of SPIONs with a high-frequency magnetic field (HFMF)—was employed against orthotopic colon cancer for the synergistic anti-tumor effect, while the tissue damage of major organs could be significantly reduced owing to the lack of systemic adsorption of SLNs into the blood circulation. Multiple studies have indicated that magnetic heat therapy is more effective than tumor chemotherapy alone in inactivating colon cancer.

### 6.3. Magnetically Controlled Release Oral Nano-Formulations

Internal or external magnetic field effects control the rate and location of drug release from MNPs, allowing precise and quantitative drug release and improved therapeutic efficacy. MNPs have many applications in biomedicine; however, their direct interactions with the biological environment may produce toxicity [68]. Biocompatible coating materials, such as liposomes, cyclic oligosaccharides, peptides, or biopolymers, are commonly used to encapsulate MNPs for functionalization and to mitigate toxicity. MNPs act as a transducer in magneto-thermal therapy, converting external electromagnetic energy into internal thermal energy, enabling controlled release of drug loaded polymer composites by disrupting chemical bonds or polymer alterations (permeability, swelling, solubility, and rigidity) [68]. Zhu [69] successfully prepared Fe_3_O_4_@OA by modifying iron oxide with pectin and oleic acid. The pectinase produced by colonic flora and the magnetic field acting on the colon degrades pectin in a specific way, resulting in nanoparticles with specifically targeted release properties in the colon. This study paves the way to synthesize multiple colon-targeting strategies and inspires advanced oral therapeutic strategies for colon cancer.

### 6.4. Oral Nano-Formulations Utilizing Magnetic Resonance Techniques

Special signals of MNPs are used in magnetic resonance imaging (MRI) to achieve tumor localization and image navigation to guide the therapeutic process. Among various diagnostic techniques, MRI is superior in imaging and image-guided drug delivery due to its excellent magnetic properties, biocompatibility, biodegradability, target modifiability, and chemical stability of MNP-loaded drugs [70,71]. MRI-based multimodal imaging for CRC has been reported with a systematic description of several commonly used imaging techniques, such as colonography, CT, CTC, MRI, and PET/CT [72]. Song et al. [50] anchored polyacrylic acid (PAA) and chitosan (CS) on Gd^3+^-doped mesoporous hydroxyapatite nanoparticles (Gd-MHAp NPs) to realize programmed drug release and MRI at tumor sites. Its association with nanoparticles allows real-time observation of drug distribution in tissues after they enter the body and is an advanced method of combining MRI with orally administered nano-chemotherapeutic drugs.

## 7. Targeted Oral Nano-Formulations

Targeted oral nano-formulations for treating CRC have recently attracted attention. Drug accumulation in tumor tissues can be improved, while toxic side effects on normal tissues can be reduced by designing and preparing nanoparticles with specific targeting properties. Targeted nanoparticles can selectively deliver drugs by targeting specific receptors or antigens on the surface of cancer cells. This targeted delivery can reduce drug decomposition and metabolism in the body while increasing drug accumulation in tumor tissues.

### 7.1. Targeting Tumor Cell Membrane Receptors

Drugs are targeted for delivery to CRC cells by recognizing and binding to specific receptors on CRC cell membranes. Several cell surface markers, such as ITGB1, EpCAM, CD44, CD133, CD24, and ALDH1, have been identified as cancer stem cell markers in CRC. However, oral formulations targeting membrane receptors must be further investigated because oral administration requires long transit via the GIT and overcoming multiple obstacles. CD44 and CD133 are the most commonly expressed receptors.

CD44 is widely expressed in intestinal inflammation and tumors. The interaction between HA and CD44 makes HA an excellent choice for delivering nanomedicines to the colon. Cheng et al. [24] prepared acid–base transformative micelles HADLA by linking HA and bile acids to form an amphiphilic conjugated polymer. Normal micelles appear fragile in gastric fluids, but HADLA targeting colonic CD44 shortens its residence time in acidic environments and allows it to be maintained at the lesion longer. These properties allow it to target the colon and exert its therapeutic effects effectively. CD133 (Prominin-1) is a transmembrane molecule identified as a cancer stem cell marker in various tumor entities, including colon cancer [73,74,75]. CD133+ cells show enhanced tumorigenicity, self-renewal pathway signaling, and metastatic properties in various cancers compared to CD133− cells [75,76,77]. Furthermore, Galizia et al. [76] demonstrated that CD133 expression correlates directly with the number of nodal metastases and subsequent tumor progression. Therefore, it is reasonable to develop new therapeutic strategies to attack CRC CSCs by targeting CD133. Zahiri et al. [77] also demonstrated that covalent attachment of the RNA aptamer of CD133 to delivery vectors effectively delivered chemotherapeutic agents to CRC cells (Figure 5). They prepared highly biocompatible nanoparticles of PCAD-DMSN@DOX, an RNA aptamer targeting a cancer stem cell (CSC) marker that covalently binds to the carboxyl group of DOX to produce CD133-PCAD-DMSN@DOX. Its specificity binds the complementary receptor on the cancer cells to increase the transportation efficiency of DOX to CD133− overexpressing HT29 tumor cells using an RNA-aptamer-targeted hybrid carrier based on dextran-coated DMSN. This potentially promising smart targeting platform may deliver drugs orally but is subject to subsequent development.

### 7.2. Targeting Intestinal Epithelial Cells

This study aimed to enhance drug absorption at the colorectal site by mimicking and utilizing the absorption mechanism of intestinal epithelial cells. This can be achieved by varying the size of nanoparticles, surface charge, and multivalent ionic interactions to increase the interaction of nano-formulations with colorectal epithelial cells. The epithelial cell adhesion molecule (EpCAM), a transmembrane glycoprotein with a molecular weight of 40 kDa, is encoded by GA-733-2 and is expressed in various epithelial tissues. It is a calcium-independent epithelial intercellular adhesion molecule in epithelial carcinogenesis and a cell surface marker for various stem and progenitor cells. Human EpCAM mutations are associated with congenital tufted enteropathy. It is also highly expressed in epithelial tumor tissues, promotes tumor proliferation, and plays a role in tumorigenesis and metastasis. It can be used as a diagnostic marker, potential prognostic marker, and target for immunotherapy of various tumors. Therefore, Liu et al. [78] developed a non-invasive anti-tumor treatment using designed anticancer protein couplings to specifically target EpCAM-ankyrin repeat protein polymers (DARPin). These results showed that a single oral drug dose cleared the HT29 colorectal tumors. In contrast, intra-tumoral injection cleared HT29 subcutaneous tumors in three doses, demonstrating that oral drugs are more readily absorbed into the systemic stream of the HT29 cancer mouse model, exert an anticancer effect on tumors in the host body, and pass via the digestive tract into the host circulation to inhibit metastatic tumors.

### 7.3. Targeting the Intestinal Mucosa

Sustained drug release and absorption in the colorectum are achieved through adhesion to the colorectal mucosa. For example, polysaccharides enable nanoparticles to adhere to the colorectum, prolong their residence time, and increase the chances of drug absorption for oral colon-targeted drug delivery. Wang et al. [79] developed a novel oral DDS that comprised active targeted nanoparticles composed of gelatin, chitosan, and Wheat Germ Agglutinin (WGA), named WGA-EF-NP. The WGA modification improved the anti-tumor activity of nanoparticles due to enhanced bio-adhesion, better bioavailability, and longer in vivo circulation time. Specifically, WGA is resistant to degradation by salivary proteases and enters the GIT smoothly via oral administration. This strategy combines the acid-resistant, controlled-release phenotype of CS-ALG with the enzyme resistance of lectin to use the colonic mucosal targeting of the particles as a strategy to overcome nonspecific adhesion to the gastric mucosa and improve the efficiency of drug delivery to the colonic site. Nanoparticles facilitate controlled drug release due to pH-dependent swelling of surface polymers, confirming their colon-specific release potential. Additionally, WGA acts as an excellent mucosal adhesion agent and binds to glycoproteins in the CM layer, exhibiting greater mucosal adhesion to intestinal tissues.

## 8. Other Types of Nano-Formulations

### 8.1. Oral Nano-Formulations for Overcoming Intestinal Barriers

Designing natural polysaccharides using micro/nanoparticle systems is an effective method for colon-targeted drug delivery. They are characterized by their high bioavailability, biocompatibility, toxicity, and low cost. Biodegradability and gastrointestinal adherence are important factors for overcoming oral delivery barriers and reaching the colonic site smoothly. Among the major polysaccharides, inulin (IN), chitosan, chitin, alginate, guar gum, starch, pectin, and HA have been widely explored for use in particulate and nano-pharmaceutical systems. Chitosan has been formulated into CSNPs with attractive features, especially in ocular and oral delivery. CSNPs are a promising strategy to overcome the low stability and bioavailability of many active ingredients [80]. Chitosan can efficiently bind to the negatively charged mucus membrane, increasing the retention time and probability of cellular uptake. Moreover, chitosan has been used as a coating agent due to its unique properties that enhance the permeation of anticancer drugs by transiently opening the tight junction between epithelial cells. Chitosan is valuable for CRC targeting due to its biodegradability via glycosidic linkage lysis by specific enzymes from the colonic microflora [19,81].

Alginate is a suitable biomaterial for designing DDS because it readily dissolves in water at room temperature and forms gels without heating and cooling cycles. It can interact with positively charged proteins in the colonic mucosa to produce hydrophilic adsorption. Colonic enzymes, like glucuronidase, can degrade it. This adherence slows the drug transit period and prolongs its residence time at the absorption site, thereby greatly improving drug bioavailability. Therefore, it is an excellent material for oral colon-targeting preparation [26,82]. A dual-targeted multi-microparticle system was designed to deliver drugs directly into the colonic domain to enhance the anticancer effects of drugs for treating CRC. Unmodified microspheres typically release drugs along the entire upper GIT, whereas encapsulated microspheres release drugs in the colonic region, where the microspheres penetrate the intestinal mucosa and reach the tumor vicinity. Finally, dual-targeted intestinal microspheres containing coupled alginate showed better colonic targeting after oral administration [83].

HA is currently the best drug delivery vehicle for intestinal diseases due to its ability to specifically bind to the receptor CD44, its transmucosal action, and its protective effect against injured mucosa. Li et al. [84] successfully synthesized a pH-responsive nano-DDS targeting the oHA receptor, oHA@ZIF-8@Oxa, thus improving the efficacy of Oxa against CRC. The oHA can compete for the binding of endogenous HA to CD44 by inhibiting the MDR and other ABC transporter protein expression, breaking the “HA-CD44” barrier in vivo, reversing the chemotherapy resistance of tumors, and penetrating the colonic mucosa more potently to play a role. Liu et al. [85] assessed the potential of CUR-encapsulated HA-zein composite nanoparticles (HZ-CUR) as an oral adjuvant for OXA-based chemotherapy in representative CRC models in mice. The specific recognition/interaction between HA and CD44 contributes to the tumor-targeting property of HA-coated nanocarriers.

In addition, Polyamidoamine dendrimer (PAMAM) is also a strategy to overcome the intestinal barrier to treat colon cancer. Its unique dendritic structure is more likely to attach to cell membranes, vesicles or mucosal surfaces, perhaps overcoming physiological barriers in the gastrointestinal tract and realizing the possibility of oral targeted drug delivery for the treatment of intestinal diseases. Its surface can be designed to connect various functional groups (COOH, COONa, NH_2_ or OH), thus reducing its surface cationic charge density and cellular toxicity. It can also be terminally attached to targeting materials (e.g., HA) to deliver drugs to specific regions [86]. Pishavar et al. [87] modified the PAMAM dendrimer molecule to enable co-delivery of chemotherapeutic agents and gene plasmids for the treatment of colon cancer. The results showed that the synthesized complexes displayed stronger anti-tumor effects than PAMAM containing only chemotherapeutic drugs or plasmids. On the other hand, treatment of mice bearing C26 colon cancer with this developed co-delivery system significantly reduced tumor growth. Therefore, this modified PAMAM can be considered as a potential vector for drug and gene co-delivery in cancer therapy.

### 8.2. Oral Nano-Formulations with Bioadhesive Properties

Bioadhesion refers to attaching charge-carrying nanoparticles to inflamed tissues via electrostatic interactions with positively charged proteins, such as eosinophils and transferrin, or negatively charged mucins [88]. Several studies have shown that the cationic surface of nanoparticles has a significant impact on the deposition pattern and therapeutic efficiency of IBDs [89]. Cationic nano-delivery systems adhere to the mucosal surface of inflamed tissues due to the interaction between positively charged nanocarriers and negatively charged intestinal mucosa [90]. Colonic mucins are negatively charged because many sulfate and silicate residues replace their carbohydrates [91]. Adhesion to the mucosa is an advantage of GI targeting because it promotes better contact with the mucosal surface, favoring cellular absorption and drug release. It also reduces nanocarrier clearance when intestinal motility increases, which is common in IBD [89]. Mucus production is also increased in patients with Crohn’s disease, thickening the mucus layer, especially in ulcerated areas, making mucus adherence a promising strategy to increase the targeting and retention of DDS in colitis.

Natural polysaccharides, such as pectin, chitosan, xylan, guar gum, and plant proteins, such as zeinolysin, have been widely used in OCDDS; they can remain intact in the upper GIT and be trapped by the mucus in the lower part of the colon for drug release [92]. Additionally, polysaccharide structural modifications or derivatives can enhance drug release behavior, site specificity, and stability [93]. Polysaccharide adhesion may facilitate prolonged contact between the drug delivery vehicle and mucosal surfaces, promoting drug absorption. Zhao et al. [94] developed sOKGM-PS-miR-31i/Cur microspheres, encapsulating both RNA inhibitors and curcumin, with a double-crosslinked carrier system that is an ideal nanomedicine rectal and oral DDS for its ultra-stable performance. The disulfide bonding interactions between the microspheres and CM layer provide ideal mucus adhesion properties. The drug strongly affected cell cycle blockade and showed optimal anti-tumor effects in animal models.

### 8.3. Intestinal Microbiome Recognition and Responsive Oral Nano-Formulations

These nano-formulations achieve selective recognition and action on intestinal microorganisms by modifying ligands or antibodies recognized by intestinal microorganisms on nanoparticle surfaces. Nanoparticles modified with specific ligands recognized by intestinal microorganisms can selectively bind to specific microorganisms in the intestinal tract, promoting beneficial intestinal flora growth and increasing drug enrichment in the colorectum.

Some studies have shown that adding prebiotics can improve the gut microbiota. Glenn Gibson and Marcel Roberfroid pioneered the concept of using prebiotics in 1995 [95]. Probiotic microorganisms selectively ferment prebiotics, producing short-chain fatty acids (SCFA), such as acetate, propionate, and butyrate [96]. Additionally, SCFA has different modes of action, including acting as an energy source for colonocytes, modulating the intestinal mucin MUC2 expression that regulates the intestinal barrier function, recognizing and regulating the intestinal microflora, and activating G protein-coupled receptors (GPCRs) signaling to modulate immune function. The gut flora ferments prebiotic-rich dietary fibers and polysaccharides to produce SCFA, which can modulate immune responses, mitigate IBD, and decrease the risk of CRC. Many studies have reported that plant-derived polysaccharides may reduce or inhibit the risk of CRC in several ways, including suppressing inflammation, modulating intestinal homeostasis, improving intestinal microbiota composition and function, and strengthening the intestinal barrier. According to Guo et al. [96], water-soluble polysaccharides (GLP) extracted from *Ganoderma lucidum* spores reduced Azoxymethane(AOM)/DSS-induced colitis and tumorigenesis. GLP administration modulated the intestinal microbiota structure and mitigated bacterial population richness and diversity after AOM/DSS treatment. In a clinical study, *Bifidobacterium bifidum*, *Lactobacillus acidophilus*, *Enterobacter cloacae*, and *Bacillus sphaericus* tablets were administered to patients with colon cancer undergoing chemotherapy. Oral probiotics improved intestinal dysbiosis and remodeled the intestinal flora diversity in patients compared to the placebo group. Meanwhile, applying probiotics can intervene in chemotherapy-induced gastrointestinal adverse reactions, and significant results have been achieved [97,98].

IN is an inexpensive, safe, and biodegradable polysaccharide derived from plants [99,100]. It is not digested or hydrolyzed by gastrointestinal saliva, intestinal mucus, and pancreatic gastrointestinal enzymes due to the presence of β-bonds in the IN structure, but rather fermented and metabolized in the colon in the presence of colonic bacteria, where CO_2_ is the metabolized form of IN. IN has become a popular colonic-targeted carrier due to this unique property, protecting the drug from the acidic environment of the stomach and upper GIT and ensuring that the IN degrades in the colon to allow controlled drug release [101]. Hou et al. [102] designed paclitaxel (PTX)-loaded oral colonic dual-targeting nanoparticles using polylactic acid-polyethyleneimine (PLA-PEI) and HA-IN. The nanoparticles were stabilized in the gastrointestinal environment with colon-targeting ability. After removing the “IN shell” by the colon bacteria, the rest of the drug exposure contributed to the inhibitory effect on colon cancer cells and achieved satisfactory therapeutic results in an in situ cancer model. Xylan and IN have similar effects. Lang et al. [103] constructed a Cap-loaded nanoparticle (SCXN) using prebiotic xylan-stearic acid conjugate. Xylan remains intact in the upper GIT and is degraded only by microorganisms in the lower GIT, making it suitable for gut-targeted drug delivery. The prebiotics produced during degradation catabolized harmful *Desulfovibro* and supported probiotic growth (*Bifidobacteria*), significantly increasing the number of beneficial bacteria and decreasing the proportion of harmful flora in the mouse gut.

A phage drug-carrying system is also an effective therapeutic strategy to modulate gut microbes in patients with bowel cancer. Phage therapy is particularly suitable to accurately remove pro-tumoral bacteria because most phages are specific to certain bacteria. Zheng et al. [104] covalently linked azodibenzocyclooctyne (DBCO)-modified IDNPs (D-IDNPs) to azide-modified phages (A-phages) to construct a phage-guided biotic–abiotic hybrid nano-system. They show that A-phages accumulated in CRC tumors in vivo, and the oral administration of the biotic–abiotic hybrid nano-system eliminated intra-tumoral *F. nucleatum*. It showed superior anti-tumor properties after oral administration in a mouse tumor model. It also showed excellent gastrointestinal and systemic tissue biosafety, with advanced bacterial recognition, in a piglet model.

## 9. Summary

Colon-targeted DDS have recently received national and international attention because they can improve the treatment of local colon diseases while minimizing systemic side effects and providing significant therapeutic benefits to patients regarding safety, efficacy, and patient compliance. Oral delivery is a better method for drug delivery. Oral formulations offer many advantages over intravenous formulations, including greater patient preference, avoidance of infusions, use of central catheters, fewer injection-related adverse events, and lower cost. A study evaluating patient preference for oral versus intravenous chemotherapy showed a clear preference for oral chemotherapy among cancer patients [105]. Overall, 9 of the 10 cancer patients preferred oral chemotherapy over intravenous chemotherapy, provided that the drug efficacy was not reduced. In a randomized crossover study in which patients with advanced CRC received either IV 5-FU or oral UFT for one cycle before switching to another treatment in the second cycle, 84% of patients preferred the oral UFT regimen due to its convenience and low toxicity. In summary, these results confirm that patients with CRC prefer oral therapy. Common reasons include convenience, flexibility, and less disruption to normal daily activities. However, when the toxicity of oral preparations exceeds that of intravenous preparations or when the therapeutic effect of oral preparations is less than that of intravenous preparations, patients with CRC prefer the latter. Additionally, intravenous anticancer therapy may cause thromboembolism due to the use of intravenous catheters, which can lead to higher costs for personnel in managing adverse events and more time being lost to additional treatment.

However, oral drugs have some disadvantages. For instance, they may be unsuitable for patients with GI obstruction or malabsorption. Additionally, treatment adherence may be reduced because oral drugs are self-administered. As mentioned earlier, the physicochemical properties of the drug, formulation, process variables, and gastrointestinal physiological factors also pose significant challenges to the successful application of colon-targeted DDS. The primary challenge is that the colon is located distally to GIT. Oral dosage forms must traverse the entire GIT to reach the colonic target site. GIT physiology is complex and influenced by food, metabolic enzymes, and intestinal bacteria, with pH, colonic fluid volume, and transport time varying greatly. These factors are barriers to reliable and efficient drug delivery into the colon. Drug solubility and dose may also be rate-limiting factors for colonic absorption due to smaller colonic fluid volume, higher viscosity, and higher pH. Additionally, colonic bacteria and enzymes may degrade drugs, rendering them ineffective. The colon consists of more than 400 different species of aerobic and anaerobic microorganisms, including *Escherichia coli* and *Clostridium difficile*, respectively [106]. These bacteria contain several hydrolytic and reductive metabolic enzymes [107]. Colonic enzymes catalyze numerous reactions, including xenobiotic metabolism, such as drug and bile acid metabolism, metabolite inactivation, and carbohydrate and protein fermentation [108]. Natural polysaccharides are commonly used in colon-targeted dosage forms as release rate control components. These polysaccharides are resistant to gastric and intestinal enzymes but are readily metabolized by anaerobic bacteria in the colon [109]. Drug metabolism by colonic enzymes can produce pharmacologically active, inactive, or harmful metabolites [110]. Formation of pharmacologically active metabolites via colonic metabolism of drugs is a common “prodrug” approach for colon-specific DDS. Finally, maintaining drug stability in the colon is a concern. Nonspecific interactions of drugs with colonic contents, such as dietary residues, intestinal secretions, mucus, or feces, may negatively affect drug stability. To ensure a balance between efficiency, target specificity, cost, and patient compliance, a combination of traditional and new approaches seems to be key in developing the OCDDS. Therefore, further research and development are necessary to realize the clinical application of OCDDS.

In conclusion, oral modes of drug delivery are an inevitable trend in the treatment of gastrointestinal disorders. With the development of different effective therapeutic strategies, combinations of new drugs and new dosage forms may help guide clinicians toward more personalized treatment of CRC patients. It is reasonable to believe that novel oral therapies may soon change the therapeutic landscape of CRC.

## Figures and Tables

**Figure 1 nanomaterials-14-00557-f001:**
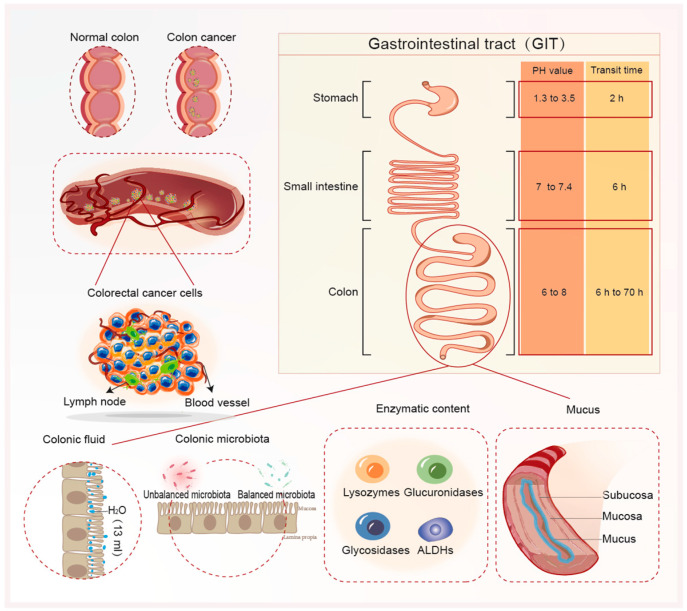
The scheme represents physiological environmental factors and major challenges for CRC therapy.

**Figure 2 nanomaterials-14-00557-f002:**
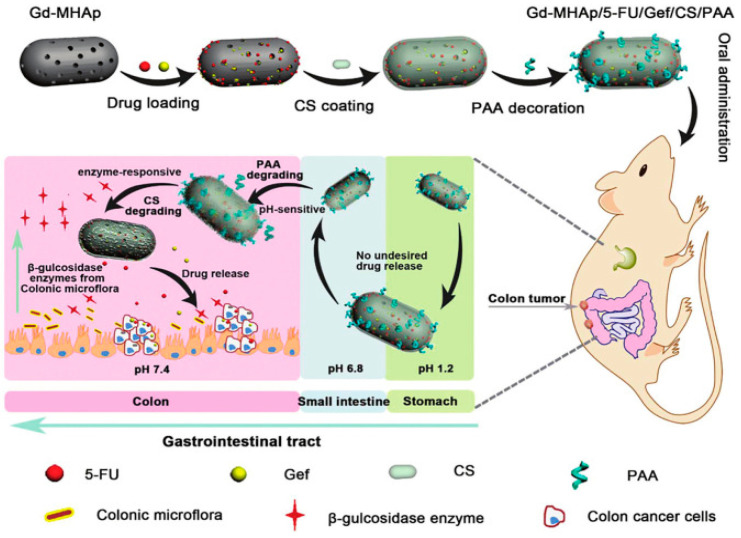
Schematic representation of the preparation of Gd-MHAPNPs and the mechanism of drug release from different physiological environments in GIT [50]. Reprinted with permission from [50].

**Figure 3 nanomaterials-14-00557-f003:**
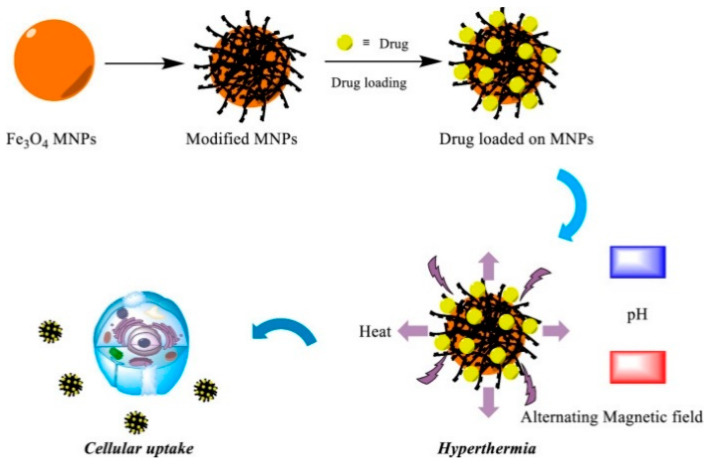
Schematic representation of the drug loading of MNPs and drug release upon application of external stimuli [65]. Author opens copyright and authorizes reprinting from [65].

**Figure 4 nanomaterials-14-00557-f004:**
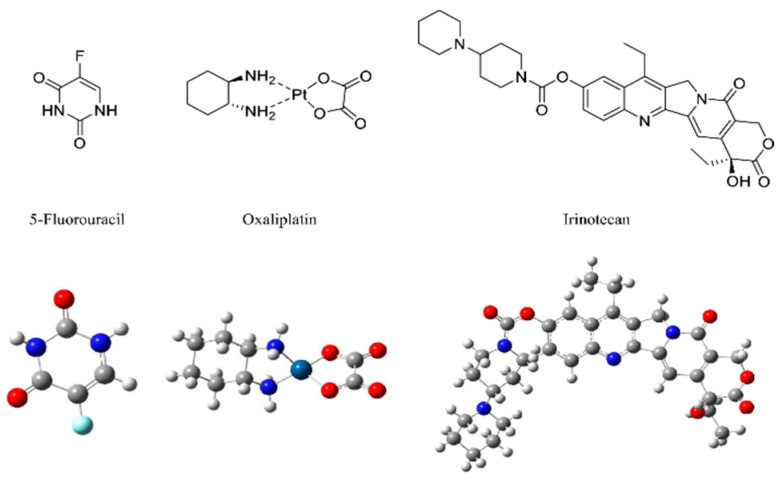
Structure of three common anticancer drugs 5-FU, oxaliplatin, and irinotecan for treating colon cancer via MNPs [65]. Author opens copyright and authorizes reprinting from [65].

**Figure 5 nanomaterials-14-00557-f005:**
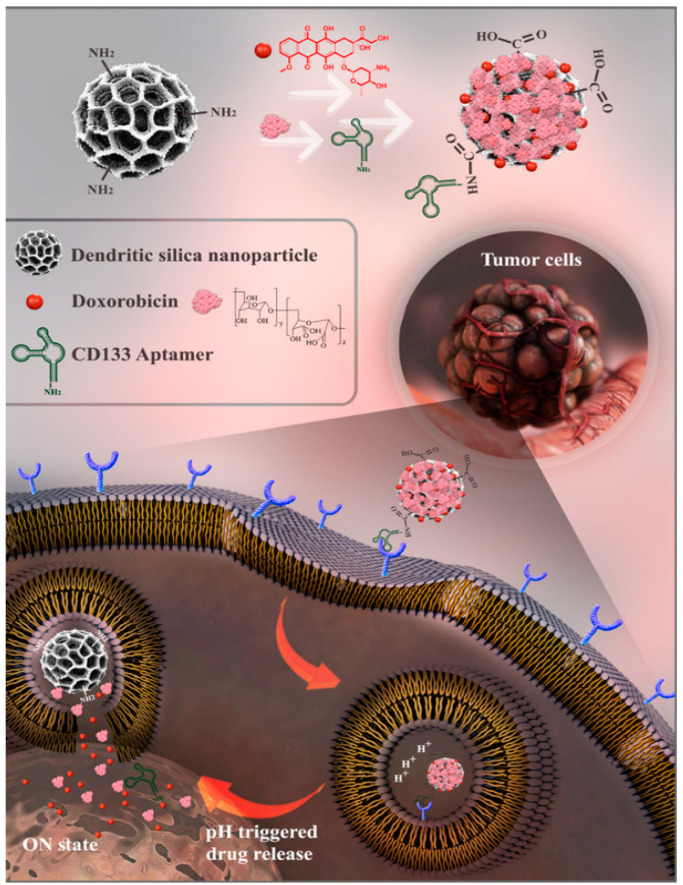
Schematic representation of CD133-targeted nanoparticle delivery mechanism [77]. Reprinted with permission from [77].

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
