# Peer review of "Current Advances of Nanomaterial-Based Oral Drug Delivery for Colorectal Cancer Treatment"

_nanomaterials, 2024, doi:10.3390/nano14070557_

Round 1
Reviewer 1 Report
Comments and Suggestions for Authors
The manuscript titled “Current Advances of Nanomaterial-Based Oral Drug Delivery for Colorectal Cancer Treatment” by Wang, N.; et al. is a Review work where the authors show the latest progress in the use of nanomaterials coming from different sources exhibiting a panoply of alternative properties. This fact allows the smart design of customized strategies against cancer diseases with special focus on the colorectar cancer. The manuscript is generally well-written and this is a topic of growing interest.
However, it exists some points that need to be addressed (please, see them below detailed point-by-point) to improve the scientifc quality of the submitted manuscript paper before this article will be consider for its publication in Nanomaterials.
1) ABSTRACT. “This study (…) novel OCDDS” (lines 14-15). Please, the full-name should be defined for the first time that one term appears in the main manuscript body text. Then, the abbreviation should be placed between brackets.
2) KEYWORDS (OPTIONAL). The authors should consider to add the full-name of the term “OCDDS” in the keyword list.
3) INTRODUCTION. “Oral colon-targeted drug delivery system (…) better cytotoxicity in CRC” (lines 39-57). Here, the authors depicted some strategies to deliver the cargo to the target colon carcinogenic cells. In this framework, it should not be neglected recent efforts devoted in this field with promising anti-inflamatory results in the mice colon histology [1].
[1] Elbassiouni, F.E.; et al. Comparative Study between Curcumin and Nanocurcumin Loaded PLGA on Colon Carcinogenesis Induced Mice. Nanomaterials 2022, 12, 324. https://doi.org/10.3390/nano12030324.
4) PHYSIOLOGICAL ENVIRONMENTAL FACTORS AND MAJOR CHALLENGES FOR CRC THERAPY. This section outlines the fundamental gastroinstentinal tract barriers. Some further information should be furnished about the gut-associated lymphoid tissue (GALT).
5) NEED AND DESIGN STRATEGIES FOR NOVEL OCDDS. “The nanoscale dimensions of the particles result in a surface area-to volume ratio ranging from 1 to 100 nm (…) from the renal system” (lines 182-186). I agree with this information provided by the authors. Nevertheless, some further information should be furnished like the nanomaterial dimensions threshold to start the visualization of aggregation effects? Does the smart design (and subsequent increase of the nanomaterial biocompatibility) minimize this detrimental effect? Some statements should be furnished in this regard.
6) “Amphiphilic lipid-based nanoparticles (…) hydrophilic surfaces (…) drug solubility” (lines 195-197). It may be advisable if the authors could mention alternative strategies to synthesize hydrophilic nanomaterials coupled with fluorescent dyes that renders efficient therapies against colorectal cancer diseases.
7) CONTROLLED RELEASE ORAL NANOFORMULATIONS. “4.2. Porous Types of Nanomaterials. (…) reaching the target site, yielding superior sutained and controlled released performance” (lines 231-245). Here, I agree with these statements provided by the authors by it should not be neglected the importance of the mechanical response of these nanomaterials [3] when they interact with the external carcinogenic cellular membrane. The proper deformation of the nanomaterials mentioned by the authors will lead the proper release of the cargo to these target cells.
[3] Magazzù, A.; et al. Investigation of Soft Matter Nanomechanics by Atomic Force Microscopy and Optical Tweezers: A Comprehensive Review. Nanomaterials 2023, 13, 963. https://doi.org/10.3390/nano13060963.
8) MICROENVIRONMENT-RESPONSIVE ORAL MATERIALS. This section perfectly outlines the most recent advances in this field (No actions are requested from the authors).
9) MAGNETIC ORAL NANO-FORMULATIONS. “An external magnetic field guides the localization of MNPs to the tumor site (…) therapeutic efficacy” (lines 363-364). Here, it is relevant to point out that the MNP magnetization strongly decays with its dimensions which is a key factor to consider for the above described statement provided by the authors.
10) TARGETED ORAL NANO-FORMULATIONS. This section is perfectly explained. No actions are requested from the authors.
11) OTHER TYPES. First, the current subsection title should be change by “Other types of nano-formulations”. Then, the authors should incorporate the opportunity to use bioconjugated dendrimers which have been successfully employed against HCT-15 human colorectal adenocarcinoma in this subsection.
12) SUMMARY. This section perfectly remarks the most relevant outcomes found by the authors in this field. The authors should add a brief statement to discuss about the future line actions to pursue this research and the open perspectives.
Author Response
Mar 10, 2024
Dear editor,
It is really a great honor for me to submit our manuscript to Nanomaterials. Please find attached files of the revised manuscript and authors’ response entitled “Current Advances of Nanomaterial-Based Oral Drug Delivery for Colorectal Cancer Treatment” by Nuoya Wang, Liqing Chen, Wei Huang, Zhonggao Gao, and Mingji Jin. This letter also contains a point-by-point reply to the peer reviewers’ comments and a summary of our responses to the editorial corrections requested. In the revised manuscript, we outlined all the changes we have made, and please do not hesitate to contact us if there’s any problem, we will reply you immediately.
It is sincerely hoped that this manuscript can be published in your journal “Nanomaterials”.
Sincerely yours,
Zhonggao Gao and Mingji Jin, Ph.D
State Key Laboratory of Bioactive Substance and Function of Natural Medicines, Institute of Materia Medica, Chinese Academy of Medical Sciences and Peking Union Medical College, Beijing 100050, China
Beijing Key Laboratory of Drug Delivery Technology and Novel Formulations, Institute of Materia Medica, Chinese Academy of Medical Sciences and Peking Union Medical College, Beijing 100050, China
E-mail: zggao@imm.ac.cn (Z. Gao); jinmingji@imm.ac.cn (M. Jin)
Point-by-point reply to the peer reviewers’ comments
Reviewer #1:
1) ABSTRACT. “This study (…) novel OCDDS” (lines 14-15). Please, the full-name should be defined for the first time that one term appears in the main manuscript body text. Then, the abbreviation should be placed between brackets.
Answer: Thank you very much for your questions! We have labeled the full name of the OCDDS and placed the abbreviation between brackets.
2) KEYWORDS (OPTIONAL). The authors should consider to add the full-name of the term “OCDDS” in the keyword list.
Answer: Thank you for your advice. I have taken your suggestion and I have added the full name of the term "OCDDS" in the keyword list.
3) INTRODUCTION. “Oral colon-targeted drug delivery system (…) better cytotoxicity in CRC” (lines 39-57). Here, the authors depicted some strategies to deliver the cargo to the target colon carcinogenic cells. In this framework, it should not be neglected recent efforts devoted in this field with promising anti-inflamatory results in the mice colon histology [1].
[1] Elbassiouni, F.E.; et al. Comparative Study between Curcumin and Nanocurcumin Loaded PLGA on Colon Carcinogenesis Induced Mice. Nanomaterials 2022, 12, 324. https://doi.org/10.3390/nano12030324.
Answer: Thank you for providing references to the excellent literature from "Nanometerials". I have quoted and paraphrased it in the introduction.
4) PHYSIOLOGICAL ENVIRONMENTAL FACTORS AND MAJOR CHALLENGES FOR CRC THERAPY. This section outlines the fundamental gastroinstentinal tract barriers. Some further information should be furnished about the gut-associated lymphoid tissue (GALT).
5) NEED AND DESIGN STRATEGIES FOR NOVEL OCDDS. “The nanoscale dimensions of the particles result in a surface area-to volume ratio ranging from 1 to 100 nm (…) from the renal system” (lines 182-186). I agree with this information provided by the authors. Nevertheless, some further information should be furnished like the nanomaterial dimensions threshold to start the visualization of aggregation effects? Does the smart design (and subsequent increase of the nanomaterial biocompatibility) minimize this detrimental effect? Some statements should be furnished in this regard.
Answer: Thanks for your suggestion. We further explain the size of nanomaterials, and other advantages of nanoparticle size control and intelligent design.
6) “Amphiphilic lipid-based nanoparticles (…) hydrophilic surfaces (…) drug solubility” (lines 195-197). It may be advisable if the authors could mention alternative strategies to synthesize hydrophilic nanomaterials coupled with fluorescent dyes that renders efficient therapies against colorectal cancer diseases.
Answer: Many thanks to the reviewers for their suggestions, we have added the relevant content.
7) CONTROLLED RELEASE ORAL NANOFORMULATIONS. “4.2. Porous Types of Nanomaterials. (…) reaching the target site, yielding superior sutained and controlled released performance” (lines 231-245). Here, I agree with these statements provided by the authors by it should not be neglected the importance of the mechanical response of these nanomaterials [3] when they interact with the external carcinogenic cellular membrane. The proper deformation of the nanomaterials mentioned by the authors will lead the proper release of the cargo to these target cells.
[3] Magazzù, A.; et al. Investigation of Soft Matter Nanomechanics by Atomic Force Microscopy and Optical Tweezers: A Comprehensive Review. Nanomaterials 2023, 13, 963. https://doi.org/10.3390/nano13060963.
8) MICROENVIRONMENT-RESPONSIVE ORAL MATERIALS. This section perfectly outlines the most recent advances in this field (No actions are requested from the authors).
Answer: We are very grateful to the reviewers for their approval and we will continue to work hard to revise our manuscript and hopefully have a chance to be accepted by "Nanometerials".
9) MAGNETIC ORAL NANO-FORMULATIONS. “An external magnetic field guides the localization of MNPs to the tumor site (…) therapeutic efficacy” (lines 363-364). Here, it is relevant to point out that the MNP magnetization strongly decays with its dimensions which is a key factor to consider for the above described statement provided by the authors.
Answer: Thanks for your suggestion. We have shown here that the magnetization strength of the MNP is size dependent.
10) TARGETED ORAL NANO-FORMULATIONS. This section is perfectly explained. No actions are requested from the authors.
Answer: We are very grateful to the reviewers for their approval and we will continue to work hard to revise our manuscript and hopefully have a chance to be accepted by "Nanometerials".
11) OTHER TYPES. First, the current subsection title should be change by “Other types of nano-formulations”. Then, the authors should incorporate the opportunity to use bioconjugated dendrimers which have been successfully employed against HCT-15 human colorectal adenocarcinoma in this subsection.
12) SUMMARY. This section perfectly remarks the most relevant outcomes found by the authors in this field. The authors should add a brief statement to discuss about the future line actions to pursue this research and the open perspectives.
Answer: Thanks for your suggestion. As a summary of the review, I have added a short statement here to discuss the benefits of conducting this study in the future as well as an open-ended perspective.
Reviewer 2 Report
Comments and Suggestions for Authors
This is a well written and comprehensive review of colonic treatments.
The Introduction and summary (section 9) give good overviews of the massive problems with trying to target the colonic cancer cells. It takes minimum of 8 hours to reach the top portion of the colon and before that any formulation must survive a tortuous route of pH, enzymes, mucus potential of debris binding etc etc. Sections 7 and 8 are ok but dont really explain how these systems avoid these issues , Some comment on the potential impracticality of these systems might be mentioned. Sectioj 6 does not give a clear idea of where magnets are and whether systemic are systemic or GI targeting. Page 15 on inulin was clear as to how a formulation might survive the GI tract to release in the colon but i was struggling to understand how other systems in sections 6 to 8 might do this.
I suggest a few cautionary comments in these sections might be appropriate or the reader will think they aremissing the point of how the systems get to the colon intact.
Specifics:
OCDDS is in abstract but not explained. intor: line 32 : DDS widely used . line 34 shoudl be targeted . page 2 line 61 and the mucosa. Page 4 line 151: and prevent foreign substances ? from what? line 154 should read: gut. If the gut symbotic micro's are starved then then :munch" Line 164 does not make sense.
page 5 line 175-198 this seems to be about systemic delivery? It needs to be put into perspective -maybe a title? : e.g. "systemic issues with DDS" .
Section 4.1 - drifts into talking about systemic related effects?
page 6 line 255: ischemia, which are associated with syn...
p7 line 271 . this looks to be systemic related info and needs to say.
figure 2 : is this acknowledged for copyright?
Section 5.2. Title says oral nano but most examples are not nano so maybe remove nano from title?
Section 5.4. line 325: "superb tissue penetration" is a big exaggeration. IT works sub cu but the intensity of radiation drops very quickly in deeper zones. - please clarify - good for sub cu- correct but not any deeper tissues such as colon.
Author Response
Mar 10, 2024
Dear editor,
It is really a great honor for me to submit our manuscript to Nanomaterials. Please find attached files of the revised manuscript and authors’ response entitled “Current Advances of Nanomaterial-Based Oral Drug Delivery for Colorectal Cancer Treatment” by Nuoya Wang, Liqing Chen, Wei Huang, Zhonggao Gao, and Mingji Jin. This letter also contains a point-by-point reply to the peer reviewers’ comments and a summary of our responses to the editorial corrections requested. In the revised manuscript, we outlined all the changes we have made, and please do not hesitate to contact us if there’s any problem, we will reply you immediately.
It is sincerely hoped that this manuscript can be published in your journal “Nanomaterials”.
Sincerely yours,
Zhonggao Gao and Mingji Jin, Ph.D
State Key Laboratory of Bioactive Substance and Function of Natural Medicines, Institute of Materia Medica, Chinese Academy of Medical Sciences and Peking Union Medical College, Beijing 100050, China
Beijing Key Laboratory of Drug Delivery Technology and Novel Formulations, Institute of Materia Medica, Chinese Academy of Medical Sciences and Peking Union Medical College, Beijing 100050, China
E-mail: zggao@imm.ac.cn (Z. Gao); jinmingji@imm.ac.cn (M. Jin)
Point-by-point reply to the peer reviewers’ comments
Reviewer #2:
1) OCDDS is in abstract but not explained. intor: line 32 : DDS widely used . line 34 shoudl be targeted . page 2 line 61 and the mucosa. Page 4 line 151: and prevent foreign substances ? from what? line 154 should read: gut. If the gut symbotic micro's are starved then then :munch" Line 164 does not make sense.
Answer: Thanks for your suggestion. I have labeled the full name of the OCDDS and placed the abbreviation between brackets. The following grammatical and logical errors have been corrected. We have corrected and detailed the causes and pathways by which the intestines become infested with bacteria.
2) page 5 line 175-198 this seems to be about systemic delivery? It needs to be put into perspective -maybe a title? : e.g. "systemic issues with DDS" .
Answer: Thank you for your suggestion. Your suggestion is so pertinent that we added the subheading " Systemic issues with DDS" to the text to make the section more logical and complete.
3) Section 4.1 - drifts into talking about systemic related effects?
Answer: Thank you for your suggestion. We have simplified the discussion of system-related impacts and added introductions relevant to the main topic, as well as citing several relevant papers
4) page 6 line 255: ischemia, which are associated with syn...
5) p7 line 271 . this looks to be systemic related info and needs to say.
Answer: Thank you for your suggestion. For systemic related information here we have added explanations and cited several relevant papers.
6) figure 2 : is this acknowledged for copyright?
Answer: Thank you for your question. This figure acknowledges copyright. The copyrights of the other figures have been marked in the revised manuscript, thank you for the reminder!
7) Section 5.2. Title says oral nano but most examples are not nano so maybe remove nano from title?
8) Section 5.4. line 325: "superb tissue penetration" is a big exaggeration. IT works sub cu but the intensity of radiation drops very quickly in deeper zones. - please clarify - good for sub cu- correct but not any deeper tissues such as colon.
Answer: Thank you for your suggestion. We also felt that the word "superb" was overstated, which was a mistake on our part, and have therefore deleted the word "superb", as explained in detail.

Reviewer 3 Report
Comments and Suggestions for Authors
The review is very comprehensive and covers all the strategies one after the other and independently of each other.
However, the writing remains very descriptive of the different results obtained in the literature, without any real analysis of these results. It lacks a critical and in-depth analysis of the different strategies and more detailed explanations of how they are supposed to work. A comparison could also provide considerable added value.
The initial description of the issues is very long and general. It could be shortened.
Author Response
Mar 10, 2024
Dear editor,
It is really a great honor for me to submit our manuscript to Nanomaterials. Please find attached files of the revised manuscript and authors’ response entitled “Current Advances of Nanomaterial-Based Oral Drug Delivery for Colorectal Cancer Treatment” by Nuoya Wang, Liqing Chen, Wei Huang, Zhonggao Gao, and Mingji Jin. This letter also contains a point-by-point reply to the peer reviewers’ comments and a summary of our responses to the editorial corrections requested. In the revised manuscript, we outlined all the changes we have made, and please do not hesitate to contact us if there’s any problem, we will reply you immediately.
It is sincerely hoped that this manuscript can be published in your journal “Nanomaterials”.
Sincerely yours,
Zhonggao Gao and Mingji Jin, Ph.D
State Key Laboratory of Bioactive Substance and Function of Natural Medicines, Institute of Materia Medica, Chinese Academy of Medical Sciences and Peking Union Medical College, Beijing 100050, China
Beijing Key Laboratory of Drug Delivery Technology and Novel Formulations, Institute of Materia Medica, Chinese Academy of Medical Sciences and Peking Union Medical College, Beijing 100050, China
E-mail: zggao@imm.ac.cn (Z. Gao); jinmingji@imm.ac.cn (M. Jin)
Point-by-point reply to the peer reviewers’ comments
Reviewer #3:
1) The review is very comprehensive and covers all the strategies one after the other and independently of each other.
However, the writing remains very descriptive of the different results obtained in the literature, without any real analysis of these results. It lacks a critical and in-depth analysis of the different strategies and more detailed explanations of how they are supposed to work. A comparison could also provide considerable added value.
The initial description of the issues is very long and general. It could be shortened.
Answer:
Thank you for your suggestions for the overview.
We have carefully considered your suggestions and made changes. We have shortened the presentation of some examples and added more detailed explanations and critical analysis. Corrections have been made to address some incorrect grammar and logic. In some strategies, we applied articles more appropriate to the topic and analyzed them in depth.
We hope you will review it again and give us your valuable comments, which will be greatly appreciated.

Round 2
Reviewer 2 Report
Comments and Suggestions for Authors
thanks for the changes